# Longitudinal Fecal Microbiota Profiles in A Cohort of Non-Hospitalized Adolescents and Young Adults with COVID-19: Associations with SARS-CoV-2 Status and Long-Term Fatigue

**DOI:** 10.3390/pathogens13110953

**Published:** 2024-10-31

**Authors:** Christine Olbjørn, Milada Hagen, Aina Elisabeth Fossum Moen, Lise Beier Havdal, Silke Lauren Sommen, Lise Lund Berven, Espen Thiis-Evensen, Tonje Stiansen-Sonerud, Joel Selvakumar, Vegard Bruun Bratholm Wyller

**Affiliations:** 1Department of Pediatric and Adolescent Medicine, Akershus University Hospital, 1478 Lørenskog, Norway; lise.beier.havdal@ahus.no (L.B.H.); silkso@ahus.no (S.L.S.); lise.lund-berven@medisin.uio.no (L.L.B.); tonje.stiansen-sonerud@ahus.no (T.S.-S.); j.p.selvakumar@medisin.uio.no (J.S.); v.b.b.wyller@medisin.uio.no (V.B.B.W.); 2Department of Nursing and Health Promotion, Oslo Metropolitan University, 0130 Oslo, Norway; milasm@oslomet.no; 3Division of Infection Control, Norwegian Institute of Public Health, 0456 Oslo, Norway; ainaelisabethfossum.moen@fhi.no; 4Institute of Clinical Medicine, University of Oslo, 0318 Oslo, Norway; 5Department of Microbiology and Infection Control, Akershus University Hospital, 1478 Lørenskog, Norway; 6Department of Gastroenterology, Rikshospitalet, Oslo University Hospital, 0372 Oslo, Norway; ethiisev@ous-hf.no; 7Department of Clinical Molecular Biology (EpiGen), University of Oslo, Akershus University Hospital, 1478 Lørenskog, Norway

**Keywords:** SARS-CoV-2, COVID-19, fatigue, post-COVID-19 condition, post-infective fatigue syndrome, microbiota, *Faecalibacterium prausnitzii*, *Sutterella wadsworthensis*

## Abstract

Adolescents most often experience mild acute COVID-19, but may still face fatigue and persistent symptoms such as post-COVID-19 condition (PCC) and post-infective fatigue syndrome (PIFS). We explored the fecal microbiota of SARS-CoV-2 positive and negative non-hospitalized adolescents and young adults (12–25 years of age) in the “Long-Term Effects of COVID-19 in Adolescents” (LoTECA) project, a longitudinal observational cohort study. With a targeted qPCR approach, the quantities of 100 fecal bacterial taxa were measured at baseline (early convalescent stage) in 145 SARS-CoV-2-positive and 32 SARS-CoV-2 negative participants and after six months in 107 of the SARS-CoV-2-positive and 19 of the SARS-CoV-2 negative participants. Results: *Faecalibacterium prausnitzii M21.2* and *Gemmiger formicilis* (both *p* < 0.001) were enriched in the SARS-CoV-2-positive participants compared to negative controls at baseline. In SARS-CoV-2-positive participants, lower baseline abundance of *Faecalibacterium prausnitzii M21/2* (*p* = 0.013) and higher abundance of *Clostridium spiroforme* (*p* = 0.006), *Sutterella wadsworthensis* (*p* < 0.001), and *Streptococcus thermophilus* (*p* = 0.039) were associated with six-month fatigue. *Sutterella wadsworthensis* and *Streptococcus thermophilus* enrichment was additionally associated with PCC in the SARS-CoV-2-positive group (*p* < 0.001 and 0.042 respectively). Conclusions: Adolescents and young adults with mild acute COVID-19 infection had increased fecal abundance of the beneficial *Faecalibacterium prausnitzii M21/2* and *Gemmiger formicilis* compared to SARS-CoV-2 negative controls in the early convalescent stage. Additionally, the abundance of both known (*Faecalibacterium prausnitzii, Streptococcus thermophilus*) and new (*Clostridium spiroforme, Sutterella wadsworthensis*) bacteria were associated with persistent symptoms such as fatigue in the COVID-19 infected group, warranting further exploration of the role of these bacteria in COVID-19 disease and PCC pathophysiology.

## 1. Introduction

Individuals affected by COVID-19 (SARS-CoV-2 infection) can exhibit a broad spectrum of post-COVID problems. Common persistent symptoms include fatigue, dyspnea, and cognitive dysfunction, and have an impact on everyday functioning [1]. Up to three-quarters of patients affected by COVID-19 describe at least one persisting symptom six months after infection, with fatigue being the most frequently reported [2,3,4]. The World Health Organization (WHO) has defined long-lasting symptoms following confirmed or suspected COVID-19 infection, with no alternative diagnosis to explain them and with a negative impact on everyday functioning, as post-COVID-19 condition (PCC) [5]. PCC exhibits significant clinical overlap with post-infective fatigue syndrome (PIFS) [6], which is defined as persistent fatigue affecting daily activities, in addition to other symptoms, following an acute infectious event [7]. Several studies indicate that the gut microbiota might contribute to COVID-19 pathogenesis and disease outcomes [8]. SARS-CoV-2 enters cells primarily through binding to angiotensin-converting enzyme 2 (ACE2) receptors, which are expressed both in the respiratory and gastrointestinal tract [9]. The amount of ACE2 receptors increases with age, and patients with an increased expression of ACE2 are more vulnerable to infection with fecal SARS-CoV-2 [10]. As children and young adults have less ACE2 receptors, this may protect them from the excess binding and penetration of SARS-CoV-2 particles on and into cells [11]. SARS-CoV-2 infection has been associated with an altered intestinal microbiota, and in adult patients, this is characterized by an enrichment of opportunistic pathogens and the depletion of beneficial commensals [11,12,13]. Several gut commensals with known immunomodulatory potential, such as *Faecalibacterium prausnitzii* (*F. prausnitzii*), *Eubacterium rectale,* and *Bifidobacteria* species, have been shown to be under-represented in adult patients infected with SARS-CoV-2 [14]. Healthy children and adolescents are usually mildly affected by SARS-CoV-2 in the acute phase of the infection and do not require hospitalization [15,16,17,18]. Children may still face post-acute illness, as individuals with mild disease seem to have the same risk of post-COVID-19 condition as hospitalized patients [19]. There is increasing evidence for an association between certain gut bacterial profiles and the severity of acute COVID-19 in adult patients [14,20,21]. In addition, gut dysbiosis seem to persist after disease resolution [22,23]. Knowledge on microbiota alterations in children and young adults during and after mild COVID-19 is limited. The aims of the present study were to investigate associations between selected gut microbiota taxa and SARS-CoV-2 status in the early convalescent stage among non-hospitalized adolescents and young adults. Further, in the SARS-CoV-2 positive cases, we aimed to assess the strengths of associations between bacterial taxa at baseline and the development of fatigue, PCC, and PIFS at six months follow-up.

## 2. Materials and Methods

The data used in the present study are part of the Long-Term Effects of COVID-19 in Adolescents (LoTECA) project, a longitudinal observational cohort study of 404 SARS-CoV-2 positive and 105 negative non-hospitalized adolescents and young adults (12–25 years of age) (see Figure 1), with a total follow-up of 12 months (Clinical Trials ID: NCT04686734). Details of the LoTECA study design have been previously published [24]. Clinical examination, laboratory testing, fecal sampling, and completion of questionnaires were performed at baseline (early convalescent stage) and at six months follow-up. Only participants who submitted a fecal sample at baseline were asked to deliver a follow-up sample at six months.

### 2.1. Study Population

From 24 December 2020, until 18 May 2021, individuals aged between 12 and 25 years old were consecutively recruited through two accredited microbiological laboratories. Individuals with laboratory-confirmed SARS-CoV-2 infection (detected by upper respiratory tract swabs followed by reverse-transcription polymerase chain reaction (RT-PCR)) were eligible for enrolment after completing quarantine (10 days). Individuals having approximately the same distribution of sex and age as the SARS-CoV-2-infected cases, but with a negative SARS-CoV-2 test from the same microbiological laboratories during the same period, were recruited as controls. Some of the controls had undergone testing due to infectious symptoms, while others were asymptomatic but had been exposed to COVID-19. The baseline assessment and inclusion took place within 28 days of a positive or negative COVID-19 test. A follow-up examination was scheduled six months after inclusion. The SARS-CoV-2 negative status was confirmed by the absence of anti-nucleocapsid antibodies at inclusion and at six months follow-up. During the first five weeks of the recruitment period, different genetic variants of the SARS-CoV-2 virus belonging to the B.1 lineage were present in this geographical area. From late-February 2021, the B.1.1.7 (alpha) variant became dominant for the remaining part of the recruitment period. Participants with gastrointestinal diseases (e.g., inflammatory bowel disease) or who were taking medications that might impact the gut microbiota (e.g., proton pump inhibitors, antibiotics) were excluded.

### 2.2. Investigational Program

Participants were summoned to a one-day investigational program at Akershus University Hospital, Norway. Participants underwent a clinical examination, spirometry, blood sampling, and fecal sampling, and completed several questionnaires, including enquiries about demographic background details and symptoms experienced during the illness episode, described in detail elsewhere [25]. The Chalder Fatigue Questionnaire (CFQ), a well-validated 11-item inventory for mental and physical fatigue [26] was used to assess fatigue at baseline and at follow-up. Each item was rated on a four-point Likert scale; in the present study, bimodal scoring (0-0-1-1) was applied, and a sum score ≥ 4 was classified as fatigue caseness in line with the current literature recommendations [27]. The application of the WHO definition of PCC (5) and the case definition for PIFS (7) at six months follow-up was performed as previously described [24]. All participants were categorized as either fatigue, PCC, and PIFS cases, or non-cases. This assessment was conducted independently by two researchers who were blinded to the participants’ initial SARS-CoV-2 status.

### 2.3. Precision Fecal Microbiota Profiling

Fecal samples from the participants were collected on Bio-Me filter cards (Bio-Me, Oslo, Norway). The cards were frozen at −80 °C until analysis. Three 6 mm discs were punched out from each sample filter card into a MagMAX Microbiome Bead Plate (ThermoFisher Scientific, Waltham, MA, USA). A Microbiome MagMAX Ultra kit (ThermoFisher Scientific, Waltham, MA, USA) was used for microbial DNA extraction. Bacterial cell walls were first disrupted using the VWR Star Beater (VWR, Radnor, PA, USA) for 2 min at 30 Hz and then using the KingFisher™ Flex (ThermoFisher Scientific, Waltham, MA, USA), following the manufacturer’s instructions with modifications as described in [28]. Purified DNA was eluted in 200 µL MagMAX Elution Buffer and quantified using Quant-iT PicoGreen dsDNA reagent (ThermoFisher Scientific, Waltham, MA, USA) and a F200 Infinite plate reader (Tecan, Männedorf, Switzerland). Bio-Me’s Precision Microbiome Profiling platform (PMP™), a target method for microbiome analysis, was used to measure the absolute quantities, genome/ng DNA, of 108 targets, including 100 different bacterial species and strains (hereby referred to as bacterial taxa; for details, see Appendix A). These taxa, known from the literature to be important for gut health and disease, were measured using quantitative PCR with reference to known concentrations of each target bacteria [29,30]. The microbiome profile was assessed using abundance of single taxa and diversity measures. Richness was assessed with PMP-alpha diversity (meaning the hits among the targeted bacteria, and the count of detected bacteria in each sample).

### 2.4. Statistical Analyses

The microbiota in fecal samples were analyzed at baseline and at six months follow-up from SARS-CoV-2 positive and SARS-CoV-2 negative participants. The data were preprocessed, and probes/variables where bacteria were detected in less than 15% of the samples, had a minimum standard deviation of less than 450 genomes/ng DNA and a maximum standard deviation less than 5000 genomes/ng DNA, or had an interquartile range (IQR) and median of 0 (0.0), were excluded, leaving 95 bacterial taxa for further analyses. Categorical variables were presented as counts and percentages. Continuous variables were presented with median and range. Crude differences between groups were assessed using Mann-Whitney U and χ^2^, or Fisher’s exact tests. Pairs of variables assessed at baseline and six months were analyzed with the Wilcoxon signed-rank test. *p*-values <0.05 were considered statistically significant. The study was regarded as exploratory; therefore, we did not adjust for multiple testing. All statistical analyses were performed using SPSS statistical software version 27 and Stata version D16.

### 2.5. Ethical Considerations

This project was conducted according to the Helsinki declaration. The project was approved by the Regional Committee for Ethics in Medical Research (ref. #203645). Written informed consent was obtained from legal guardians for participants < 16 years of age, or from the participants themselves when older than 16 years of age.

## 3. Results

At baseline, fecal samples were collected from 145 (36%) out of 404 SARS-CoV-2 positive and from 32 (30%) out of 105 SARS-CoV-2 negative participants. At six months, among those who provided baseline samples, 107 (74%) of the 145 SARS-CoV-2 positive and 19 (59%) of the 32 SARS-CoV-2 negative participants submitted a follow-up fecal sample. After excluding three participants with other diseases and/or medications, and six participants whose samples did not pass laboratory quality control steps, we analyzed the fecal microbiota of 136 SARS-CoV-2 positive and 32 SARS-CoV-2 negative participants at baseline, and of 102 SARS-CoV-2 positive and 17 SARS-CoV-2 negative participants at six months follow-up (Figure 1 and Table 1).

### 3.1. Fecal Bacterial Abundances and SARS-CoV-2 Status at Baseline

Principal component analysis did not reveal any bacterial clusters separating the SARS-CoV-2 positive and negative participants (Appendix A), nor was there a difference in bacterial richness between the two groups of participants at baseline (Appendix A). However, the abundances of 14 bacterial taxa were statistically different between samples from SARS-CoV-2 positive compared to SARS-CoV-2 negative participants (Table 2), with an enrichment of six taxa, including *F. prausnitzii M21.2* (Figure 2A), *Gemmiger formicilis* (*G. formicilis*) (Figure 2B), *Holdemanella biformis* (*H. biformis*), *Flavonifractor plauti* (*F. plauti*), and *Eggerthella lenta* (*E. lenta*), in the SARS-CoV-2 positive participants. Eight species had significantly reduced bacterial abundances (Table 2), including *Phocaeicola massiliensis (P. massiliensis)* and *Bifidobacterium longum (B. longum)*.

### 3.2. Fecal Bacterial Abundances and SARS-CoV-2 Status at Six Months

In fecal samples collected at six months follow-up, seven bacterial taxa revealed diverging abundances between SARS-CoV-2 positive participants and controls (Table 2). These taxa were different from those found separating the two groups at baseline.

### 3.3. Associations Between Bacterial Taxa at Baseline and Fatigue, PCC and PIFS Among SARS-CoV-2 Positive Participants at Six Months

Higher baseline abundance of *Sutterella wadsworthensis* (*S. wadsworthensis)* was associated with fatigue caseness at inclusion (approximately half of the SARS-CoV-2 positive participants) (Figure 3A). The 46 (35%) SARS-CoV-2 positive participants classified as fatigue cases at six months follow-up had lower abundances of *F. prausnitzii M21/2* (Figure 3A) and *Ruminococcus bicirculans* (*R. bicirculans*), and higher abundances of five bacterial taxa including *Clostridium spiroforme* (*C. spiroforme*) (Figure 4B), *S. wadsworthensis* (Figure 3B), and *Streptococcus thermophilus* (S. *thermophilus)* at baseline compared to non-cases (Table 3). The 43% of SARS-CoV-2 positive participants with PCC (56/130) at six months had higher abundances of *S. wadsworthensis* (Figure 3C) and *S. thermophilus* at baseline compared to those without PCC (Table 3). The baseline abundance of *S. wadsworthensis* was also higher in the 15 (11.5%) SARS-CoV-2 positive participants with PIFS at six months, although this association failed to reach statistical significance (*p* = 0.09; Figure 3D). There were no bacterial taxa at baseline associated with PIFS status (15 (11.5%) SARS-CoV-2 positive participants) at six months.

## 4. Discussion

The main finding of the present study was an enrichment of the beneficial short chain fatty acid (SCFA) producers *F. prausnitzii M21.2* and *G. formicilis* in the SARS-CoV-2 positive adolescents and young adults compared to SARS-CoV-2 negative controls at baseline. Six months post-infection this enrichment was no longer evident. Furthermore, in the SARS-CoV-2 positive group, lower baseline levels of *F. prausnitzii M21/2* and *R. bicirculans* and a higher abundance of *C. spiroforme, S. wadsworthensis,* and *S. thermophilus* were associated with fatigue at six months. Additionally, enrichment of *S. wadsworthensis* and *S. thermophilus* at baseline was associated with PCC.

### 4.1. SARS-CoV-2 Status

The increased abundances of *F. prausnitzii* and *G. formicilis* in the SARS-CoV-2 positive participants at baseline conflicts with results from adult COVID-19 cohorts where these SCFA producing and beneficial species were reported as being depleted [14,31,32]. However, the enrichment of *F. prausnitzii* is consistent with gut microbiota studies on children with COVID-19 [33]. Our findings support the hypothesis that *F. prausnitzii* serves as a hallmark species enriched in pediatric COVID-19 disease, and that it may play a protective role against severe disease manifestations and outcomes [33,34]. Another player in maintaining a healthy gut microbiota, the anti-inflammatory bacterium *H. biformis*, was also found to be more abundant in our SARS-CoV-2 positive participants compared to controls [35,36]. However, although *F. prausnitzii*, *G. formicilis*, and *H. biformis* were enriched in our SARS-CoV-2 positive participants, several other commensal bacteria were depleted in comparison with controls. *P. massiliensis* (formerly *Bacteroides massiliensis*) [37], a beneficial, mucine-degrading member of the gut microbiota with the ability to utilize glycans [38], was observed at a decreased level in our COVID-19 patients, alongside a depletion of *B. longum. Bifidobacteria* interact with human immune cells and modulate specific pathways and may decrease proinflammatory cytokines and restore intestinal barrier integrity [39]. Depletion of *Bifidobacteria* has been reported in adult patients with COVID-19 [14], as well as in pediatric patients [34]. *F. plautii*, enriched in our SARS-CoV-2 positive participants, has in previous studies been found to be increased in pediatric inflammatory bowel disease (IBD) and in irritable bowel syndrome (IBS) patients [40,41]. Similarly, we found an increased abundance of *E. lenta*, a bacterium associated with autoimmunity and previously found enriched in the gut microbiota of children with multisystem inflammatory syndrome (MIS-C) [42]. Taken together, we find evidence of both anti- and pro-inflammatory differences in the microbiota of SARS-CoV-2 positive versus SARS-CoV-2 negative controls at baseline following mild COVID-19 disease in adolescents and young adults.

### 4.2. SARS-CoV-2 Infection and Severity

One hypothesis regarding the less severe course of COVID-19 disease in children compared with adults is that their gastrointestinal microbiota differs from that of adults [43], and that pre-infection differences in the gut microbiota, such as enrichment of SCFA producers and associated immune responses, may determine COVID-19 disease presentation, severity and outcome [17,44]. The SARS-CoV-2 positive adolescents and young adults in our study were not hospitalized and did not have severe manifestations of COVID-19. It is possible that the enrichment of the SCFA producers *F. prausnitzii* and *G. formicilis* contributed to the relatively mild disease course. Lower levels of *F. prausnitzii* observed in populations at risk for severe COVID-19 disease, such as the elderly, and obese and diabetic populations [45], further support this theory. In line with these results, we found an association of fatigue caseness at six months with lower baseline abundances of *F. prausnitzii M21/2* and *R. bicirculans* [46], the latter being a beneficial SCFA producer and one of the most abundant members of the healthy gut microbiota. These findings support the hypothesis surrounding the post COVID-19 gut microbiome possibly having an impaired capacity to synthesize SCFAs, and that low SCFA levels may contribute to persistence of symptoms such as fatigue and PCC [47]. Similarly, we found an association between lower abundance of *Bifidobacterium longum* and *Bifidobacterium animalis* with SARS- CoV-2 status, and lower abundance of *Bifidobacterium angulatum* with fatigue caseness. Together with *F. prausnitzii* [48,49,50], there is evidence for an inverse association between disease severity and *Bifidobacteria* abundance [45]. *Bifidobacteria* have previously been linked to anti-infection and anti-viral activity, as well as being a valuable player in regulating the host immune system [51]. It is possible that one of the reasons that we generally observe a less severe disease course of COVID-19 in children is due to their high abundance of *Bifidobacteria*. It would be of interest to see whether probiotic treatment with *Bifidobacteria* strains could be beneficial in the prevention and treatment of COVID-19.

An interesting finding in the present study is the enrichment of the Gram-negative, microaerophilic bacterium *S. wadsworthensis* at baseline in the SARS-CoV-2 positive participants with fatigue at six months, as well as in those fulfilling the criteria for PCC. This bacterial species is a potential pathogen associated with pediatric IBD [52], as well as pediatric obesity [53]. *S. wadsworthensis* has not been implicated in PCC or PIFS before, but the ability of *Sutterella* species to adhere to intestinal epithelial cells could indicate an immunomodulatory role [54]. Similarly, *S. thermophilus* was enriched at baseline in SARS-CoV-2 patients with fatigue and PCC at six months. In line with this, Li et al. found *S. thermophilus* to be enriched in COVID-19 patients compared to healthy controls, and that this species correlated with severity of the disease [55].

Another novel association discovered in the present study was the higher abundance of *C. spiroforme* in participants with fatigue following COVID-19 infection compared to those without fatigue post-infection. *C. spiroforme* has recently been linked to the development and progression of colorectal cancer in [56]. Another study from 2024 found that the presence of this species in the feces may aid in the diagnosis of IBD [57], indicating that *C. spiroforme* may act as a pathogen.

The bacterial taxa showing deviating abundances in participants with PCC and PIFS were not the same in the SARS-CoV-2 positive as in the SARS-CoV-2 negative participants, although the SARS-CoV-2 negative participants had a similarly high symptom load as the participants who tested positive for SARS-CoV-2. It is not possible to determine whether the differences in bacterial taxa levels observed between our SARS-CoV-2 positive participants and controls preceded or followed infection with COVID-19. While we cannot attribute a causal role of bacterial taxa in the pathogenesis of SARS-CoV-2 infection in our study, our data support a potential role of the gut microbiome in modulating the disease course, as hypothesized earlier by Rochi et al. [58].

### 4.3. Strengths and Limitations

One of the strengths of this study is the well-defined group of non-hospitalized symptomatic young individuals with COVID-19 and a comparable SARS-CoV-2 negative control group. All participants were subjected to rigorous and systematic examinations, and they were from the same catchment area, of similar socio-economic status, and were not treated with medications during their COVID-19 illness. SARS-CoV-2 antibody testing was conducted both at inclusion and at six months follow-up, and participants displaying antibodies indicative of prior COVID-19 infection were excluded as SARS-CoV-2 negative controls.

Our study has several limitations. The sample size of the study, particularly regarding SARS-CoV-2 negative participants, was small, with only one in three of the LoTECA enrolled participants being willing to deliver fecal samples, rendering them eligible for this sub-study. Due to the small sample size and no correction for multiple testing, our findings with deviating species abundance in fatigue and PCC could be coincidental and must be interpreted with caution. In addition, there is a potential risk of self-selection bias, as those who joined the study and provided a fecal sample may not be representative for the targeted population of children/adolescents. As species and strains to be detected were selected a priori, we may have missed novel bacteria with previously unknown beneficial/harmful effects. The sample material used in the study was microbial DNA; hence, the results represent active, dormant, and dead microbes. Furthermore, we had no influence over or knowledge of the participants’ diets and lifestyles, which are known to influence gut microbiota and could be an important confounder in our analyses.

## 5. Conclusions

Adolescents and young adults with mild acute COVID-19 infection had increased fecal abundance of the beneficial SCFA producers *F. prausnitzii and G. formicilis* species compared to SARS-CoV-2 negative participants in the early convalescent stage. Additionally, the abundance of both known (*F. Prausnitzii*, *S. thermophilus*) and new (*C. spiroforme* and *S. wadsworthensis*) bacteria were associated with persistent symptoms such as fatigue in the COVID-19 infected group, warranting further exploration of the role of these bacteria in COVID-19 and PCC pathophysiology.

## Figures and Tables

**Figure 1 pathogens-13-00953-f001:**
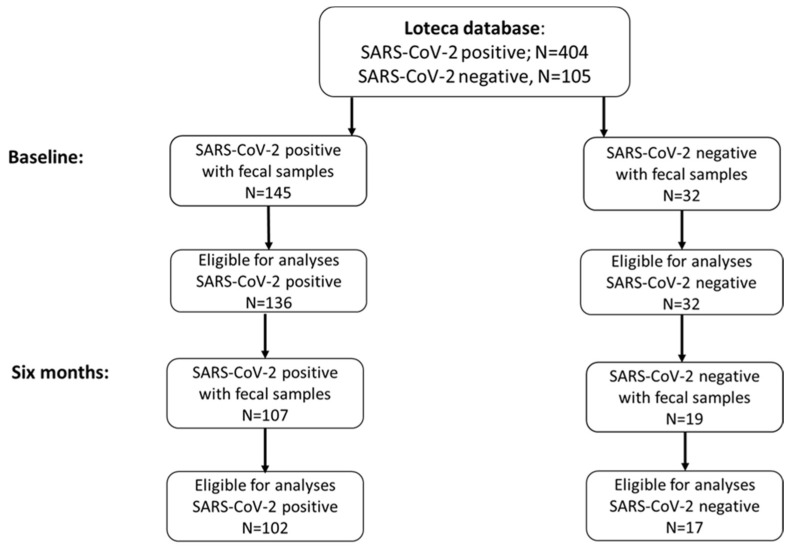
Flowchart.

**Figure 2 pathogens-13-00953-f002:**
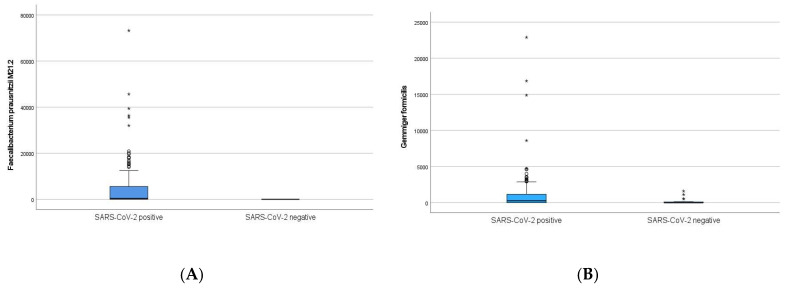
Boxplots showing abundance (genomes/ng DNA) of (**A**) *Faecalibacterium prausnitzii M21.2* and (**B**) *Gemmiger formicilis* in SARS-CoV-2 positive participants compared to SARS-CoV-2 negative controls at baseline.

**Figure 3 pathogens-13-00953-f003:**
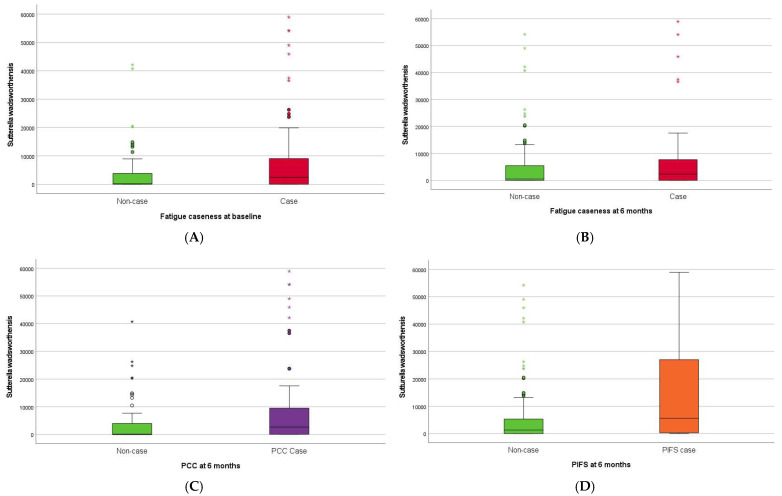
Boxplots showing baseline abundance (genomes/ng DNA) of *Sutterella wadsworthensis* in SARS-CoV-2 positive participants with (**A**) fatigue case versus non-case at baseline, (**B**) fatigue case versus non-case at six months, (**C**) PCC (post-COVID-19 condition) case versus non-case at six months, and (**D**) PIFS (post-infective fatigue syndrome) case versus non-case at six months.

**Figure 4 pathogens-13-00953-f004:**
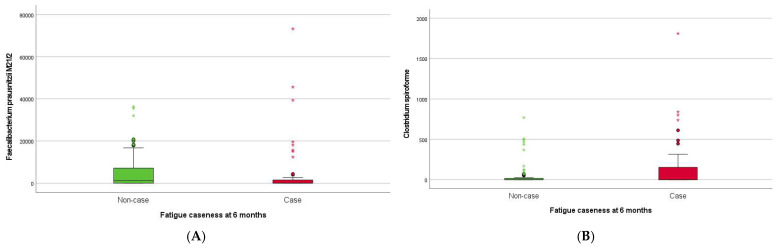
Boxplots showing baseline abundance (genomes/ng DNA) of (**A**) *Faecalibacterium prausnitzii M21.2* and (**B**) *Clostridium spiroforme* in SARS-CoV-2 positive participants with fatigue case compared to non-case at six months.

**Table 1 pathogens-13-00953-t001:** Characteristics of SARS-CoV-2 positive and SARS-CoV-2 negative participants including fatigue caseness (positive CFQ bimodal score) at baseline and six months follow-up, and post-COVID-19 condition (PCC) and post-infective fatigue syndrome (PIFS) at six months follow-up. Abbreviations: IQR, interquartile range; BMI, body mass index; CFQ, Chalder fatigue questionnaire.

	Baseline		Six Months Follow-Up	
Characteristics	SARS-CoV-2 Positive (n = 136)	SARS-CoV-2 Negative (n = 32)	SARS-CoV-2 Positive (n = 102)	SARS-CoV-2 Negative (n = 17)
Female, n (%)	78 (57)	19 (59)	46 (45)	12 (71))
Age median [IQR]	17.0 [14,15,16,17,18,19,20,21]	16.5 [15,16,17,18,19,20]	17.0 [14–22.5]	18.0 [16,17,18,19,20,21]
BMI median [IQR]	22.4 [19.9–25.3]	21.8 [20.2–24.4]	22.3 [19.8–25.3]	21.6 [20–23.8]
CFQ fatigue caseness, n (%)	71 (51)	14 (45)	36 (35)	4 (24)
PCC caseness, n (%)			42 (42)	7 (41)
PIFS caseness, n (%)			12 (12)	1 (6)

**Table 2 pathogens-13-00953-t002:** Bacterial taxa with an increased (↑) or a decreased (↓) abundance (ng/genome DNA) in SARS-CoV-2 positive participants compared to SARS-CoV-2 negative controls at baseline and at six months follow-up. Abbreviations: ns (not significant).

Bacterial Taxa with Deviating Abundance in SARS-CoV-2 Positive Compared to SARS-CoV-2 Negative	↑ or ↓ in SARS-CoV-2 Positive at Baseline	↑ or ↓ in SARS-CoV-2 Positive at Six Months	*p*-Value
*Faecalibacterium prausnitzii M21.2*	↑	ns	<0.001
*Gemmiger formicilis*	↑	ns	<0.001
*Gordonibacter pamelaeae*	↑	ns	0.003
*Holdemanella biformis*	↑	ns	0.005
*Flavonifractor plautii*	↑	ns	0.031
*Phocaeicola massiliensis*	↓	ns	0.012
*Holdemanella filiformis*	↓	ns	0.005
*Eggerthella lenta*	↑	ns	0.021
*Odoribacter splanchnicus*	↓	ns	0.028
*Alistipes shahii*	↓	ns	0.032
*Alistipes finegoldii*	↓	ns	0.035
*Bacteroides uniformis*	↓	ns	0.035
*Clostridium citroniae*	↓	ns	0.045
*Bifidobacterium longum*	↓	ns	0.046
*Bifidobacterium animalis*	ns	↓	0.007
*Faecalibacterium prausnitzii CNCM4575*	ns	↓	0.014
*Streptococcus anginosus*	ns	↓	0.017
*Bacteroides stercoris*	ns	↓	0.019
*Clostridium nexile*	ns	↓	0.038
*Parabacterium merdae*	ns	↓	0.042
*Eubacterium eligens*	ns	↑	0.048

**Table 3 pathogens-13-00953-t003:** Bacterial taxa with a significantly increased (↑) or decreased (↓) abundance at baseline in SARS-CoV-2 positive participants with fatigue, PCC (post-COVID-19 condition), and PIFS (post-infective fatigue syndrome) at six months follow-up compared to SARS-CoV-2 positive participants without fatigue, PCC, and PIFS. Abbreviations: ns (not significant).

Bacterial Taxa with Deviating Abundance in Case Compared to No Case	SARS-CoV-2 Positive Participants (n = 130)			
	Baseline ↑ or ↓ (*p* Value) in Case n (%)	Six-Months Follow-Up ↑ or ↓ (*p* Value) in Case n (%)		
	Fatigue 68 (52%)	Fatigue 46 (53%)	PCC 56 (43%)	PIFS 15 (12%)
*Bacteroides thetaiomicron*	↑ (0.026)	ns	ns	ns
*Sutterella wadsworthensis*	↑ (0.047)	↑ (<0.047)	↑ (<0.001)	ns
*Alistipes putredenis*	↑ (0.049)	ns	ns	ns
*Bifidobact angulatum*	↓ (0.014)	ns	ns	ns
*Phocoeicola massiliensis*	↓(0.014)	ns	ns	ns
*Bacteroides stercoris*	↓ (0.025)	ns	ns	ns
*Clostridium spiroforme*	ns	↑ (0.006)	ns	ns
*Faecalibacterium prausnitzii CNCM75*	ns	↑ (0.024)	ns	ns
*Streptococcus thermophilus*	ns	↑ (0.039)	↑ (0.042)	ns
*Roseburia intestinalis*	ns	↑ (0.046)	ns	ns
*Faecalibacterium prausnitzii M21/2*	ns	↓ (0.013)	ns	ns
*Ruminococcus bicirculans*	ns	↓(0.045)	ns	ns

## Data Availability

The raw datasets generated and analyzed during the current study are not publicly available in order to protect participant confidentiality. The data files are stored by Akershus University Hospital, Lørenskog, Norway, on a server dedicated to research. The security follows the rules given by The Norwegian Data Protection Authority, P.O. Box 8177 Dep. NO-0034 Oslo, Norway. The data are available on request to the authors.

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
