# Peer review of "Longitudinal Fecal Microbiota Profiles in A Cohort of Non-Hospitalized Adolescents and Young Adults with COVID-19: Associations with SARS-CoV-2 Status and Long-Term Fatigue"

_pathogens, 2024, doi:10.3390/pathogens13110953_

Round 1
Reviewer 1 Report
Comments and Suggestions for Authors
Good work!
Comments:
1. Why are only specific bacteria targeted to be studied? What are the criteria for their selection?
2. What were the variants of the positive participants of the study? Was this issue enlightened? Has any association between gut microbiota and SARS-CoV-2 variants been found?
3. The gut microbiome probably protects children regarding the severity of COVID-19, however it is common knowledge that the numbers of ACE-2 molecules on/in tissues increase with the years of age, and this normal function practically plays a main role in children’s clinical outcomes: children have smaller ACE-2 numbers leading to more mild clinical manifestations since smaller numbers of viral particles can bind on cells and penetrate. To be commented, so that no confusing impressions are created regarding the protective role of gut microbiome.
L. 175-180: Why are these bacteria particularly mentioned among others? To be explained.
Author Response
Response to Reviewer 1.
Thank you very much for taking the time to review this manuscript. Please find the detailed responses below and the corresponding revisions/corrections highlighted in the re-submitted files.
Reviewer’s Evaluation:
Does the introduction provide sufficient background and include all relevant references? Yes
Is the research design appropriate? Can be improved
Are the methods adequately described? Yes
Are the results clearly presented? Yes
Are the conclusions supported by the results? Can be improved
Response to Reviewer’s Evaluation:
Regarding the research design: The research design was planned right after the outbreak of the SARS-CoV-2 epidemic. In hindsight we would have planned it differently, especially regarding the study of the gut microbiota. We would have looked at the different SARS-CoV-2 variants separately, we would have tried to include more participants in the SARS- CoV-2 negative control group and we would have liked to do a more in-depth analysis of the microbiota.
Regarding whether the conclusions are supported by the results: To improve this, we have edited conclusions in the manuscript, page 10, starting from line 354, and conclusions in the abstract, page 1, starting from line 30, in the following way: Adolescents and young adults with mild acute COVID-19 infection had increased fecal abundance of the SCFAs producers F. prausnitzii and G. formicilis species compared to SARS-CoV-2 negative controls in the early convalescent stage. Additionally, the abundance of both known (F. Prausnitzii, S. thermophilus) and new (C. spiroforme and S. wadsworthensis) bacteria were associated with persistent symptoms such as fatigue in the COVID-19 infected group, warranting further exploration of the role of these bacteria in COVID-19 and PCC pathophysiology.
Point-by-point response to Comments
Comments 1: Why are only specific bacteria targeted to be studied? What are the criteria for their selection?
Response 1: Thank you for pointing this out. We had access to Bio-Me’s Precision Microbiome Profiling platform (PMP™) and were able to pay for these tests. PMP™ is a cheaper and faster method for microbiota analysis, as it is a qPCR method measuring the absolute quantities, genome/ng DNA of the targets included in the test, and not a sequencing method. Bio-Me is updating their targets regularly. At the time when we measured the gut microbiota of our participants, the PMP™ contained the 108 targets that are listed in the supplementary file, as they had been found to be important in health and disease. In the manuscript this is pointed out on page 3, in lines 144-146: “These taxa, known from the literature to be important for gut health and disease, were measured using quantitative PCR with reference to known concentrations of each target bacteria (29,30)”.
Comments 2: What were the variants of the positive participants of the study? Was this issue enlightened? Has any association between gut microbiota and SARS-CoV-2 variants been found?
Response 2: Thank you for this important question. As stated in the manuscript, page 3, line 108-111: “During the first five weeks of the recruitment period, different genetic variants of the SARS-CoV-2 virus belonging to the B.1 lineage were present in this geographical area. From late-February 2021, the B.1.1.7 (alpha) variant became dominant for the remaining part of the recruitment period.” we did not group our positive participants according to which variant they had, as this would have made the groups for comparison quite small. We are unaware of studies where they compare different SARS-CoV-2 variants with the gut microbiota, but as the variants evolve this would be very interesting to investigate.
Comments 3. The gut microbiome probably protects children regarding the severity of COVID-19, however it is common knowledge that the numbers of ACE-2 molecules on/in tissues increase with the years of age, and this normal function practically plays a main role in children’s clinical outcomes: children have smaller ACE-2 numbers leading to more mild clinical manifestations since smaller numbers of viral particles can bind on cells and penetrate. To be commented, so that no confusing impressions are created regarding the protective role of gut microbiome.
Response 3: We agree with this excellent comment. Accordingly, we have added a paragraph to the manuscript to emphasize this point. In the introduction, page number, 2, starting from line 56: “SARS-CoV-2 enters cells primarily through binding to angiotensin-converting enzyme 2 (ACE2) receptors which are expressed both in the respiratory and gastrointestinal tract (9). Patients with gut microbiome dysbiosis have an increased expression of ACE2 on their mucosal cells (10), making them more vulnerable to an infection by fecal SARS-CoV-2. The amount of ACE2 receptors increases with age, and patients with an increased expression of ACE2 are more vulnerable to an infection with fecal SARS-CoV-2 (10). As children and young adults have less ACE2 receptors, this may protect them from an excess binding and penetration of SARS-CoV-2 particles on and into cells (11). SARS-CoV-2 infection has been associated with an altered intestinal microbiota, and in adult patients this is characterized by an enrichment of opportunistic pathogens and depletion of beneficial commensals (11-13).
Comments 4 L. 175-180: Why are these bacteria particularly mentioned among others? To be explained.
Response 4: Thank you for pointing this out. These were the bacteria that had a statistically significantly different abundance. All bacteria are mentioned in Table 2, in the text I highlight five out of six with an enrichment because I mention them later in the discussion. I mention two species in the discussion with a reduced abundance, I have omitted the other two from the text that I do not discuss further, Alistipes shahii and Odoribacter splanchnicus. Now in the results section of the manuscript, page 5, starting from line 192 it says: “However, the abundances of 14 bacterial taxa were statistically different between samples from SARS-CoV-2 positive compared to SARS-CoV-2 negative participants (Table 2), with an enrichment of six taxa, including F. prausnitzii M21.2 (Figure 1A), Gemmiger formicilis (G. formicilis) (Figure 1B), Holdemanella biformis (H. biformis), Flavonifractor plauti (F. plauti) and Eggerthella lenta (E. lenta) in the SARS-CoV-2 positive participants. Eight species had significantly reduced bacterial abundances (Table 2), including Phocaeicola massiliensis (P. massiliensis) and Bifidobacterium longum (B. longum).”
- Response to Comments on the Quality of English Language
English language fine. No issues detected.
New reference: Silva MG, Falcoff NL, Corradi GR, Di Camillo N, Seguel RF, Tabaj GC, et al. Effect of age on human ACE2 and ACE2-expressing alveolar type II cells levels. Pediatric research. 2023;93(4):948-52.

Reviewer 2 Report
Comments and Suggestions for Authors
The overall manuscript is interesting and quite informative, however, I suggest to assess a few aspects better
1 The authors stated that: Patients with gut microbiome dysbiosis have an increased expression of ACE2 on their mucosal cells (10), making them more vulnerable to an infection by fecal SARS-CoV-2. My critique is: Did you check ACE2 presence in both groups of affected and non-affected patients?
2 Please specify the total of participants in both the study and control group since the beginning (Abstract section) because it is quite confusing
3 The most important is this question. Bifidobacteria as probiotic bacteria strains belonging to the genus Bifidobacterium colonize the gastro-intestinal tract of humans and animals at the time of birth, and they are found in young and adult individuals in large numbers. Moreover, they can interact with the development of enteric infections by producing antimicrobial metabolites. Have you tested the presence of any of these bifidobacteria, to better understand the extensiveness of infection and long-term fatigue due to COVID-19? How the SARS-CoV-2 my have a better chance of spreading with or without them?
Comments on the Quality of English Language
English needs just a minor adjustment
Author Response
Response to Reviewer 2.
Thank you very much for taking the time to review this manuscript. Please find the detailed responses below and the corresponding revisions/corrections highlighted in the re-submitted files.
Reviewer’s Evaluation:
Does the introduction provide sufficient background and include all relevant references? Yes
Is the research design appropriate? Can be improved
Are the methods adequately described? Can be improved
Are the results clearly presented? Can be improved
Are the conclusions supported by the results? Can be improved
The overall manuscript is interesting and quite informative, however, I suggest to assess a few aspects better.
Response to Reviewer’s Evaluation:
Regarding the research design: The research design was planned right after the outbreak of the SARS-CoV-2 epidemic. In hindsight we would have planned it differently, especially regarding the study of the gut microbiota. We would have looked at the different SARS-CoV-2 variants separately, we would have tried to include more participants in the SARS- CoV-2 negative control group and we would have liked to do a more in-depth analysis of the microbiota.
Regarding the methods: As this is a part of the Long-Term Effects of COVID-19 in Adolescents (LoTECA) project, LOTECA, the methods have been described in detail in the publication by Selvakumar J, Havdal LB, Drevvatne M, et al. Prevalence and Characteristics Associated With Post-COVID-19 Condition Among Nonhospitalized Adolescents and Young Adults. JAMA Netw Open. 2023;6(3):e235763. This is referred to in the manuscript in the Material and method section, page 2, line 115: “The data in the present study are part of the Long-Term Effects of COVID-19 in Adolescents (LoTECA) project, a longitudinal observational cohort study of 404 SARS-CoV-2 positive and 105 negative non-hospitalized adolescents and young adults (12-25 years of age) (see Flowchart) , with a total follow-up of 12 months (Clinical Trials ID: NCT04686734). Details of the LoTECA study design are previously published (24).
Additionally, in the material and method section, page2, line 84 we refer to an added a flowchart, page 4, line 180 to better visualize the number of participants and fecal samples:
In the abstract, page 19-24 we have modified the methods section, and it now states:
We explored the fecal microbiota of SARS-CoV-2 positive and negative non-hospitalized adolescents and young adults (12-25 years of age) in the “Long-Term Effects of COVID-19 in Adolescents” (LoTECA) project, a longitudinal observational cohort study. With a targeted qPCR approach, the quantities of 100 fecal bacterial taxa were measured at baseline (early convalescent stage) in 145 SARS-CoV-2-positive and 32 SARS-CoV-2 negative participants and after six months in 107 of the SARS-CoV-2-positive and 19 of the SARS-CoV-2 negative participants.
Regarding the results: We have added a flowchart, page 4, starting from line 180, to better visualize the participants with eligible fecal samples for analyses.
Regarding whether the conclusions are supported by the results: To improve this, we have edited the conclusions section in the manuscript in the following way, page 10, starting from line 353 : “Adolescents and young adults with mild acute COVID-19 infection had increased fecal abundance of the beneficial SCFA producers F. prausnitzii and G. formicilis species compared to SARS-CoV-2 negative participants controls in the early convalescent stage. Additionally, the abundance of both known (F. Prausnitzii, S. thermophilus) and new (C. spiroforme and S. wadsworthensis) bacteria were associated with persistent symptoms such as fatigue in the COVID-19 infected group, warranting further exploration of the role of these bacteria in COVID-19 and PCC pathophysiology”.
Point-by-point response to Comments
Comments 1: The authors stated that: Patients with gut microbiome dysbiosis have an increased expression of ACE2 on their mucosal cells (10), making them more vulnerable to an infection by fecal SARS-CoV-2. My critique is: Did you check ACE2 presence in both groups of affected and non-affected
Response 1: Thank you for pointing out this issue. Unfortunately, we did not check the ACE2 presence in the SARS-CoV-2 negative vs positive group. As this was an exploratory study we did not think of this when we planned the study. It would have been interesting to do so. The only thing we can say is that our participants were age matched and since ACE2 receptor presence is age dependent we believe that their presence should be relatively equal.
To deal with this issue we have modified the section of the manuscript: We have deleted reference nr 10, Changed order of reference nr 11 to 10, added new reference Silva 2023. In the introduction, page 2, line 58-62: it now says: “SARS-CoV-2 enters cells primarily through binding to angiotensin-converting enzyme 2 (ACE2) receptors which are expressed both in the respiratory and gastrointestinal tract (9). The amount of ACE2 receptors increase with age, and patients with an increased expression of ACE2 are more vulnerable to an infection with fecal SARS-CoV-2 (10). As children and young adults have less ACE2 receptors, this may partly explain why they often display a mild disease course following infection with SARS-CoV-2 (11). “
Reference nr 11: Silva M, 2023 Effect of age on human ACE2 and ACE2-expressing alveolar type II cells levels. M. G. Silva, N. L. Falcoff, G. R. Corradi, N. Di Camillo, R. F. Seguel, G. C. Tabaj, et al.
Pediatr Res 2023 Vol. 93 Issue 4 Pages 948-952.
Comments 2: Please specify the total of participants in both the study and control group since the beginning (Abstract section) because it is quite confusing
Response 2: We agree with this comment and have added the following in the abstract, page 1, starting from line 19:
We explored the fecal microbiota of SARS-CoV-2 positive and negative non-hospitalized adolescents and young adults (12-25 years of age) in the “Long-Term Effects of COVID-19 in Adolescents” (LoTECA) project, a longitudinal observational cohort study with a total follow-up of 12 months. With a targeted qPCR approach, the quantities of 100 fecal bacterial taxa were measured at baseline (early convalescent stage) in 145 SARS-CoV-2-positive and 32 SARS-CoV-2 negative participants and after six months in 107 of the SARS-CoV-2-positive and 19 of the SARS-CoV-2 negative participants.
We have also added the flowchart, page 4, starting from line 181.
Comments 3. The most important is this question. Bifidobacteria as probiotic bacteria strains belonging to the genus Bifidobacterium colonize the gastro-intestinal tract of humans and animals at the time of birth, and they are found in young and adult individuals in large numbers. Moreover, they can interact with the development of enteric infections by producing antimicrobial metabolites. Have you tested the presence of any of these bifidobacteria, to better understand the extensiveness of infection and long-term fatigue due to COVID-19? How the SARS-CoV-2 my have a better chance of spreading with or without them?
Response 3: Thank you for this comment. Among the 108 targets in the Bio-Me PMPTM test we find several bifidobacteria strains: Bifidobacterium adolescentis, Bifidobacterium angulatum, Bifidobacterium animalis subsp. Lactis, Bifidobacterium bifidum, Bifidobacterium catenulatum, Bifidobacterium longum, Bifidobacterium longum subsp. Longum and Bifidobacterium pseudocatenulatum. At baseline, only Bifidobacterium longum had a significantly differently abundance in the SARS-CoV-2 positive compared to the SARS-CoV-2 negative, as mentioned in Table 2 and in the Discussion, page 8 under 4.1. SARS-CoV-2 status, line 269-273: “was decreased in our COVID-19 patients, alongside a depletion of B. longum. Bifidobacteria interact with human immune cells and modulate specific pathways and may decrease proinflammatory cytokines and restore intestinal barrier integrity (39). Depletion of Bifidobacteria are reported in adult patients with COVID-19 (14) as well as in pediatric patients (34).
When measured at 6 months, this Bifidobacterium strain no longer had a significantly differently abundance, but at 6 months, Bifidobacterium animalis was less abundant in the SARS-CoV-2 participants, as listed in Table 2.
The presence of Bifidobacterium angulatum was lower in the SARS-CoV-2 positive with fatigue case at baseline compared to the SARS-CoV-2 positive without fatigue.
To elaborate on the role of Bifidobacteria, we have added the following in the discussion, under the section 4.2. SARS-CoV-2 infection and severity, page number 9, starting from line 299-309: “Similarly, we found an association between lower abundance of Bifidobacterium longum and Bifidobacterium animalis with SARS- CoV-2 status and lower abundance of Bifidobacterium angulatum with fatigue caseness. Depletion of Bifidobacteria species have been reported both in adults (43) and in children (34) with COVID-19. Together with F. prausnitzii (40-42) there is evidence for an inverse association between disease severity and Bifidobacteria abundance (43). Bifidobacteria have previously been linked to anti-infection and anti-virus activity as well as being a valuable player in regulating the host immune system (new reference: J. Chen et al). Maybe one of the reasons for a less severe disease course of COVID-19 in children is due to their high abundance of Bifidobacteria. It would be of interest to see whether probiotic treatment with Bifidobacteria strains could be beneficial in the prevention and treatment of COVID-19.
- Response to Comments on the Quality of English Language: English needs just a minor adjustment.
Thank you for your comment. We have done some minor adjustments to the written English and hope this has improved the manuscript.
New reference:
- Chen, X. Chen and C. L. Ho. Recent Development of Probiotic Bifidobacteria for Treating Human Diseases. Front Bioeng Biotechnol 2021 Vol. 9 Pages 770248

Round 2
Reviewer 2 Report
Comments and Suggestions for Authors
The paper completely lacks an immunological part that may give a wider scientific foundation
For instance, it has been established that Faecalibacterium is abundant in normal populations and has been shown to have protective benefits on digestive health while also enhancing the immune system, metabolism, and gut barrier of the host. Faecalibacterium can potentially be a therapeutic target in diseases connected to the microbiota, such as immunological disorders and cancer immunotherapy. It would be interesting to hear your opinion on this since your study asserted almost completely contrary information.
In addition, the paper needs to have an immunology part regarding the possible presence of SNPs and the crosstalk between microbiota, immune system, and COVID-19 infection... (Balzanelli et al)
Comments on the Quality of English Language
Minor English editing needed
Round 3
Reviewer 2 Report
Comments and Suggestions for Authors
In my opinion the article can be accepted